# A Perspective on Reversibility in Controlled Polymerization Systems: Recent Progress and New Opportunities

**DOI:** 10.3390/molecules23112870

**Published:** 2018-11-03

**Authors:** Houliang Tang, Yi Luan, Lu Yang, Hao Sun

**Affiliations:** 1School of Materials Science and Engineering, University of Science and Technology Beijing, 30 Xueyuan Road, Haidian District, Beijing 100083, China; houliangt@smu.edu; 2Department of Chemistry, Southern Methodist University, 3215 Daniel Avenue, Dallas, TX 75275, USA; 3Department of Chemistry, University of Florida, PO Box 117200, Gainesville, FL 32611-7200, USA; yanglulucia@chem.ufl.edu

**Keywords:** controlled polymerization, reversible polymerization, sustainable polymers

## Abstract

The field of controlled polymerization is growing and evolving at unprecedented rates, facilitating polymer scientists to engineer the structure and property of polymer materials for a variety of applications. However, the lack of degradability, particularly in vinyl polymers, is a general concern not only for environmental sustainability, but also for biomedical applications. In recent years, there has been a significant effort to develop reversible polymerization approaches in those well-established controlled polymerization systems. Reversible polymerization typically involves two steps, including (i) forward polymerization, which converts small monomers into macromolecule; and (ii) depolymerization, which is capable of regenerating original monomers. Furthermore, recycled monomers can be repolymerized into new polymers. In this perspective, we highlight recent developments of reversible polymerization in those controlled polymerization systems and offer insight into the promise and utility of reversible polymerization systems. More importantly, the current challenges and future directions to solve those problems are discussed. We hope this perspective can serve as an “initiator” to promote continuing innovations in this fairly new area.

## 1. Introduction

The fundamental concept of reversibility has been widely utilized to drive the development of new polymeric materials, which can display distinct but reversible change in properties upon receiving a stimulus [1,2]. In light of this, a library of reversible materials based on polymers have recently been achieved, including self-healing materials bearing reversible-covalent linkages [3,4,5,6,7,8], recyclable materials such as vitrimers [9,10,11], polymer networks enabling reversible sol-gel transitions [12,13,14], architecture-transformable polymers [15], and covalent or metal organic frameworks harnessing reversible bonds [16,17,18,19,20,21]. Despite the tremendous success in the aforementioned polymer systems, little attention has been paid to achieving reversible polymerizations. Compared with self-immolative polymers, which can only undergo one-way depolymerization, a reversible polymerization typically features reversible transformations between polymers and original monomers [22]. Indeed, step-growth polymerizations relying on reversible-covalent chemistry, particularly Diels–Alder chemistry, have provided an approach to reversible polymers that favors forward polymerization at room temperature and tends to depolymerize at high temperatures (i.e., 120 °C) [23]. However, those polymers prepared by step-growth polymerizations typically have very broad molecular weight distributions and small molecular weights, limiting their potentials in certain applications that require precise polymer chain length, complicated architectures, and high molecular weights [24,25,26,27,28,29].

In the last two decades, the rapid advent of controlled and living polymerizations has offered polymer scientists a powerful synthetic toolbox for accessing polymers with predetermined molecular weights, well-defined architectures, and narrow distributions in molecular weights [30,31,32,33,34,35]. In addition, the excellent tolerance of functional groups in controlled polymerization systems has enabled us to achieve advanced polymer materials with desired functions and properties [36,37,38,39,40,41]. In general, controlled polymerization techniques can be categorized into three common classes. The first one involves ring-opening polymerization (ROP) of cyclized monomers (e.g., caprolactones, lactides, and *N*-carboxyanhydrides) in the presence of a nucleophilic initiator and a catalyst (metal or organic) [42,43,44,45,46,47]. The second category highlights the well-developed applications of controlled anionic/cationic polymerizations, which have led to industrial production of thermoplastic elastomers [30]. The last class focuses on reversible-deactivation radical polymerization (RDRP) methodologies, which are capable of polymerizing vinyl monomers, such as acrylates, methacrylates, and styrene in a controlled manner [48,49,50,51]. To date, three mainstream RDRP techniques, including atom-transfer radical polymerization (ATRP) [52,53], reversible addition–fragmentation transfer (RAFT) polymerization [54,55,56,57], and nitroxide-mediated polymerization have been developed [58]. Among them, ATRP and RAFT are receiving the most attention, due to their unique advantages, such as mild polymerization conditions, broad monomer scope, and ease of end-group functionalization [59]. These controlled polymerization systems are enjoying tremendous success in producing polymers that accommodate both industrial use (supported by ROP and anionic/cationic polymerizations) and academic research. However, the studies related to depolymerization are scarce, because depolymerization was typically considered as a side reaction that would lessen the performance—e.g., the mechanical properties of polymer materials [60]. While this is true in the pursuit of maximizing lifetime or long-term stability of polymer materials, serious pollution problems have arisen from the lack of degradability in commercial polymer plastics under common conditions [61]. Sustainable polymers, such as degradable polyesters deriving from biomass resources, represent a promising platform for environmental remedy. However, those polymers currently suffer from high manufacturing cost and low mechanical properties compared to vinyl polymer-based materials from petroleum resources. Moreover, the degradation of these sustainable polyesters is typically one-way, resulting in non-polymerizable fragments that are not useful for the regeneration of new materials. To fully achieve sustainability in polymer materials, recent attentions have been shifted to the development of new methods for depolymerizing polymers back into original monomers under accessible conditions [60,62,63,64,65,66,67,68,69,70,71]. These regenerated monomers can be recycled and further repolymerized to obtain new polymer materials. From this perspective, we aim to first critically assess the state-of-the-art toolbox for achieving reversible polymerization in controlled polymerization systems (Scheme 1). Recent examples on reversible polymerizations will be classified according to their controlled polymerization mechanisms, i.e., ROP and RDRP (Scheme 1 and Table 1). In addition, we believe it is important to assess the current challenges of reversible polymerizations that can trigger polymer chemists to solve this issue. Finally, potential applications deriving from this fairly new concept will be predicted and discussed.

## 2. Reversible Polymerization in Ring-Opening Polymerization Systems

We begin our exploration of reversible polymerization approaches in ROP systems [63,64,66,67,68,69,70,71]. As one of the most popular controlled polymerization strategies, ROP of cyclic monomers has emerged as a useful synthetic route to prepare technologically interesting polymers with desirable architectures and specific properties. In particular, nucleophile-initiated ROP in the presence of metallic or organocatalysts allows the polymerization to proceed in a controlled manner, affording polymers with pre-determinable molecular weights and narrow molecular weight distributions. Applicable monomers include lactones, lactides, cyclic carbonates, *N*- or *O*-carboxyanhydrides, and cyclooligosiloxanes, among others [43]. To date, well-defined polymers produced from those monomers have attracted significant interest in both academic research and industry [43,72,73] 

Although many of the ROP polymers are comprised of hydrolysable ester linkages in their backbones, which can cause the polymers to degrade into oligomers or possibly small molecules, it is still challenging to completely convert the polymers back to the original cyclized monomers. In 2014, Albertsson and coworkers demonstrated their pioneering work on ring-closing depolymerization to obtain a functional six-membered cyclic carbonate monomer, 2-allyloxylmethyl-2-ethyltrimethlene carbonate (AOMEC), from its oligomeric form [63]. The synthesis of AOMEC was performed in a one-pot reaction, involving the oligomerization of trimethylolpropane allyl ether, diethyl carbonate, and NaH, followed by an in-situ anionic depolymerization. It was observed that the depolymerization can occur beyond the ceiling temperature of the polyester at a certain polymer/monomer concentration, suggesting the reversible nature of the polymerization of AOMEC. However, there was still room to improve, since the oligomers were synthesized simply by condensation reaction rather than ROP, and the degree of polymerization was only from 1 to 7.

Inspired by this pioneering work, Albertsson’s team continued the study on AOMEC and first demonstrated the reversible ROP of this six-membered cyclic carbonate monomer in 2016 [64]. It was concluded that the equilibrium between controlled ROP of AOMEC and controlled ring-closing depolymerization (RCDP) of poly(AOMEC) were dictated by various reaction parameters, such as monomer concentration, reaction temperature, and even solvents. In their approach, AOMEC was polymerized by ROP in the presence of an organocatalyst—that is, 1,8-diazabicyclo(5.4.0)undec-7-ene (DBU) either in bulk or in different solvents (Figure 1a). Various temperatures were used to evaluate the equilibrium conversion for calculating the ceiling temperature at certain polymerization conditions. According to the thermodynamic principle, ROP of AOMEC was favored at a temperature lower than the ceiling temperature. As the reaction temperature was higher than the ceiling temperature, RCDP of poly(AOMEC) dominated. It is worth noting that the molecular weights increased/decreased linearly as a function of monomer conversion in the cases of both ROP and RCDP (Figure 1b). Moreover, the molecular weight distribution of all the evolving polymers remained as low as 1.1, which further verified the exceptional control over the course of both polymerization and depolymerization.

As one of the best suitable biomass-derived compounds to replace petroleum-derived chemicals, γ-butyrolactone (γ-BL) and its polymer PγBL have great potential as sustainable materials [68,70]. However, γ-BL has been commonly considered to be “non-polymerizable” due to its low strain energy, which has thereby rendered the ROP of γ-BL extremely challenging. Starting from 2016, Chen et al. has conducted extensive research on ROP of γ-BL, and have now developed several exciting synthetic methodologies to achieving PγBL by ROP. In agreement with Albertsson’s viewpoint on the thermodynamic basis, Chen anticipated that reaction conditions should be modulated to achieve successful ROP, including a much lower reaction temperature than the ceiling temperature, a high initial monomer concentration, and most importantly, a robust catalyst. In their first work associated with γ-BL, the ROP of γ-BL was carried out at −40 °C in the presence of a lanthanide (Ln)-based coordination polymerization catalyst [68]. The polymerization was capable of affording PγBL with M_n_ as high as 30,000 g mol^−1^ and up to 90% monomer conversion. In addition, the polymer topology (e.g., linear or cyclic) can also be controlled by varying the initiator structure and the feeding ratio of raw materials (Figure 2). From the sustainable perspective, a quantitative depolymerization of PγBL was realized by heating the purified polymers for 1 h at higher temperatures (i.e., 220 or 300 °C) than the ceiling temperature. Interestingly, the PγBL that was dissolved in appropriate solvents depolymerized much more rapidly in the presence of an organocatalyst (e.g., 1,5,7-triazabicyclo[4.4.0]dec-5-ene) or metal catalyst (e.g., La[N(SiMe_3_)_2_]_3_), even at room temperature.

In parallel, Chen and coworkers also discovered a metal-free ROP strategy to produce PγBL, using a super-basic organocatalyst with abbreviation tert-Bu-P_4_ (Figure 3a) [71]. It should be noted that the catalyst itself was able to initiate the ROP of γ-BL at −40 °C, by abstracting protons from γ-BL to form highly reactive enolate species (Figure 3b). However, the monomer conversion was limited up to 30.4%, due to the possible interference of [tert-Bu-P4H]^+^ and an anionic dimer. Furthermore, the ROP performance was greatly enhanced when tert-Bu-P_4_ was added, along with a suitable alcohol serving as the initiator (e.g., BnOH). With the help of the alcoholic initiator, the monomer conversion reached as high as ca. 90%, and the corresponding PγBL possessed a molecular weight of 26,700 g mol^−1^. Notably, the PγBL prepared by this organocatalyzed ROP was completely recyclable, and can depolymerize back to γ-BL upon heating at 260 °C for 1 h.

Encouraged by the success of preparing fully-recyclable PγBL via ROP with either metal (La, Y) or organocatalysts, Chen’s group proceeded to apply this unique polymerization process to an enriched variety of monomers. ROP of α-Methylene-γ-butyrolactone (MBL), a small molecule derived from biomass and regarded as a potential alternative to the petroleum-based MMA, was subsequently investigated (Figure 4) [69]. Since the monomer comprises a non-strained five-membered lactone and a highly reactive exocyclic C=C double bond, many knee-jerk studies were exclusively focused on traditional vinyl addition polymerization (VAP) [74,75]. In Chen’s work, the lanthanide (Ln)-based coordination polymerization catalyst was utilized, leading to an unsaturated polyester P(MBL) with M_n_ up to 21,000 g mol^−1^ through the ROP process. Remarkably, by adjusting the reaction conditions, such as the catalyst (La)/initiator (ROH) ratio and temperature, three pathways of the MBL polymerization can be realized independently, including conventional VAP, ROP, and crosslinking polymerization. As was foreseeable, only the polymers resulted from the ROP pathway were fully recyclable.

Despite the notable achievement in lanthanide (La, Y) or superbase (tert-Bu-P_4_) catalyzed ROP of γ-BL and its derivatives, the undesirable low temperature (i.e., −40 °C) to implement the process still remained as one of biggest hurdles for industry use. Moreover, the as-synthesized polymers suffered from limited thermostability and crystallinity. Hence, the exploration of new materials with both superior properties and energy economy are still in demand. Recently, Chen’s group proposed a γ-BL derivative (abbreviated as 3,4-T6GBL), with a cyclohexyl ring transfused to the five-membered lactone at the α and β positions [70]. This monomer can be polymerized by using the coordinative insertion ROP catalysts, producing linear or cyclic polymers with high molecular weights at room temperature (Figure 5a). Following this finding, Chen et al. extended the scope of ROP of another γ-BL derivative (4,5-T6GBL), where the cyclohexyl ring was fused at the β and γ (or 4,5) positions of the BL ring, giving rise to linear/cyclic polymers at room temperature (Figure 5b) [67]. A controlled polymerization behavior was observed as the molecular weights of evolving polymers, which increased linearly with the monomer conversions. As expected, in both studies the resulting polymers possess enhanced thermostability and could be quantitatively recycled back to their original building monomers by either thermolysis or chemolysis.

Very recently, Hoye et al. described the synthesis of a novel substituted polyvalerolactone from a malic acid derived monomer, 4-carbomethoxyvalerolactone (CMVL) [66]. In their work, this six-membered ring monomer was blended with a diol (1,4-benzenedimethanol) and an organic acid (diphenyl phosphate) at an ambient temperature, finally forming a semicrystalline material with a molar mass up to 71,000 g mol^−1^. Notably, the resulting polymer can be either depolymerized back into its original precursor monomer or degraded into acrylate-type analogues (Figure 6a). The former process was catalyzed by tin octanoate (Sn(Oct)_2_), providing 87% monomer recovery; while the latter was promoted by DBU with a comparable yield. In particular, a substantial kinetics study showed that CMVL was polymerized smoothly and reached ca. 90% conversion after 15 h (Figure 6b). It also revealed that nearly all of the initiator was consumed within 20 min. This “fast initiation” can be regarded as one of the characteristic behaviors of controlled polymerization systems.

## 3. Reversible Polymerization in Reversible-Deactivation Radical Polymerization Systems

Although reversible polymerizations in ROP systems have shown great promise in next-generation sustainable polymer materials, the major market for commodity polymers is still occupied by vinyl polymers, due to their low cost in manufacturing. Vinyl polymers are derived from petroleum, a non-renewable resource. Therefore, the recycling of used vinyl polymer materials such as plastics has immense merits, not only in global waste reduction, but also for petroleum sustainability [76]. In this section, recent examples in depolymerization of vinyl polymers derived from RDRP systems will be discussed. We hope those timely developments will prompt more innovative thinking with regard to plastic recycling via a reversible polymerization approach.

In 2012, Zhu et al. reported the first example of reversible polymerizations in an RDRP system [62]. In their pioneering work, vinyl polymerizations of several acrylamide monomers, including *N*-isopropylacrylamide (NIPAM) and *N*,*N*-dimethylacrylamide (DMA), were successfully achieved in the presence of CuCl and tris(2-dimethylaminoethyl)amine (Me6TREN) (Figure 7a). Very intriguingly, they unexpectedly observed a phenomenon of depolymerization when radical inhibitors like 2,2,6,6-tetramethylpiperidinooxy (TEMPO) or 1,4-benzoquinone (BQ) were added to the ongoing polymerization system, with the initial purpose of terminating radical polymerizations. To further elucidate the role of the copper catalyst in the depolymerization process, a control experiment with regard to a conventional radical polymerization, using 2,2′-azobisisobutyronitrile (AIBN) as radical initiator, was carried out in the absence of a copper catalyst and ligands. TEMPO was added during the conventional radical polymerization, resulting in only the termination of polymerization, without any noticeable depolymerization. Those results unequivocally verified that a copper catalyst is essential in depolymerization. Therefore, a depolymerization mechanism based on β-alkyl elimination from the copper (II) coordination center was proposed (Figure 7b).

In a similar demonstration of reversible polymerization mediated by a copper catalyst, Haddleton and coworkers were able to prepare well-defined polyacrylamides and polyacrylates through aqueous copper-mediated radical polymerization in the presence of dissolved CO_2_ (Figure 8a) [65]. In the case of a NIPAM monomer, the forward polymerization adopted rapid reaction kinetics, achieving full monomer conversion within 10 min. Thereafter, a significant in-situ depolymerization occurred to an extent of 52%, and thereby led to the regeneration of the NIPAM monomer, which was systemically confirmed by proton nuclear magnetic resonance (NMR), gel permeation chromatography (GPC), and electron ionization-mass spectroscopy. Importantly, this recycled NIPAM can be repolymerized upon deoxygenation of the resulting solutions, illustrating the reversibility of the polymerization (Figure 8b). Furthermore, the scope of reversible copper-mediated polymerization was extended to *N*-hydroxyethyl acrylamide (HEAm) and 2-hydroxyethyl acrylate (HEA), demonstrating the versatility of this system. However, it should be noted that the mechanism of depolymerization, as well as the role of CO_2_ in the depolymerization process, was not identified in their study.

In the aforementioned RDRP systems (Vide supra), the depolymerization phenomenon was only observed during the course of the polymerization. However, it is arguably more interesting from the materials point of view if one can depolymerize a polymer post-synthesis or after the manufacturing process. In very recent work described by Gramlich, a set of brush polymers consisting of oligo-ethylene glycol or oligo-dimethylsiloxane side chains were prepared by traditional RAFT polymerization in the presence of AIBN at 70 °C (Figure 9a) [60]. After polymerization, those polymers were thoroughly purified by repeated precipitations, to ensure the removal of residual monomers and initiators. Upon purification, thermally-induced depolymerization of the as-synthesized polymer was conducted in dilute dioxane solutions, leading to the regeneration of vinyl monomers until reaching the monomer’s inherent equilibrium monomer concentration. Importantly, the residual polymers exhibited high chain-end fidelity by retaining the trithiocarbonate moiety after depolymerization, allowing for further reinitiation and repolymerization via a RAFT mechanism (Figure 9b,c).

## 4. Closing Remarks

The current success in reversible polymerizations has enabled us to think about many new possibilities in future polymer science. However, many challenges still remained to be addressed, hampering the further translation of this new concept into real-world applications. One apparent hurdle is how to achieve good control over the depolymerization process. While all the examples covered in this perspective are related to controlled polymerizations, allowing for a predictable degree of forward polymerizations, little information was provided to reveal the kinetics of the depolymerization process, especially those involved in controlled radical polymerization. Indeed, previous literature has placed much focus on the start and endpoint of depolymerization (in other word, the highest degree of depolymerization). Notwithstanding, kinetic study will shed more light on the fundamental mechanisms of reversible polymerizations, and if one can predetermine and control the degree of depolymerization by changing several reaction parameters, such as time, temperature, catalyst/initiator loading, polymer/monomer concentrations, among others. Moreover, the ability to tune the depolymerization rate under normal conditions is expected to open the door to many interesting applications (in addition to sustainable materials)—for example, self-healable materials, and sustained release systems, which require slow and controllable depolymerization. It is worth noting that the concurrent depolymerization approaches are typically associated with harsh conditions (e.g., high temperatures, metal catalysts), significantly impeding the translation of this concept into biomedical uses. In light of this, the continuing pursuit of new depolymerization methodologies that can be implemented under mild and physiological conditions will be important towards bio-related applications.

Another challenge stems from the relatively low efficiency in depolymerizations, particularly those in RDRP systems. In comparison with ROP-based depolymerization systems, which mainly rely on breaking weak polyester backbones, the energy input necessary for reversing vinyl polymer backbones (i.e., carbon–carbon single bonds) back to vinyl monomers is considerably higher. To our best knowledge, the highest reported depolymerization conversion in RDRP systems was only 71% when N-hydroxyethyl acrylamide was involved in copper(0)-mediated reversible polymerizations [65]. In the RAFT mediated depolymerization approach, only 30% of monomers can be regenerated after heating the diluted RAFT polymer solutions at 70 °C under a vacuum for several days [60]. From the viewpoint of potential industrial applications, insufficient depolymerization could dramatically increase the cost deriving from separating regenerated monomers from residual polymers. Therefore, we envision that more attention will be paid to detailed mechanism study and the rational design of depolymerization systems, with the goal of achieving high depolymerization efficiency (such as the effort for lowering equilibrium monomer concentration). Moreover, we believe that mathematical tools, such as modeling and simulations of reversible polymerizations, should play a key role in prediction of the dynamics, final products, and optimal conditions in reversible polymerizations [77]. While the concept of reversible polymerizations is still in its infant stage, it is anticipated that the future development in this area will not only deepen our understanding of fundamental depolymerization mechanisms, but also promote many new opportunities and applications in polymer science and engineering.

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
