# Peer review of "A Perspective on Reversibility in Controlled Polymerization Systems: Recent Progress and New Opportunities"

_molecules, 2018, doi:10.3390/molecules23112870_

Reviewer 1 Report

This manuscript submitted by Tang et al. reviewed the current research status of reversible controlled polymerization. They summarized some pioneering work in this area from Albertsson, Chen, Gramlich and others, with focusing on ROP, ATRP and RAFT polymerization.  The authors had a broad and deep view of this area with comprehensive literature review. This manuscript was well written with proper discussions of current status, opportunities, and future challenges. I believe this review will inspire more research interests and attract attention in polymers and materials community. I recommend publishing this review in Molecules with minor revisions.

1. Line 50-51: The authors considered the controlled polymerization can be categorized into two common classes: ROP and CRP. However, controlled anionic/cationic polymerizations were also important and well developed controlled polymerization techniques as well as used in producing industrial thermoplastics elastomers like SBS, SIS etc. So I suggest the authors to add controlled anionic/cationic polymerizations as the 3rd controlled polymerizations for a general description;

2. Line 61-62: “These controlled polymerization systems are enjoying tremendous success in producing polymers accommodated for both industrial use and high-value added applications.” This claim is true for ROP with wide applications in industrial point. However, the controlled radical polymerizations (CRP), like RAFT and ATRP, mainly focus in academia while less success in industrial applications with very few commercial polymers available.  So it’s not proper to claim as this;

3. Figure 1a: The polymer structures were not correct. The polymers should be polycarbonates with repeat unit-[O-C(O)-O-R]-. However, Figure 1 has one more O in the backbone of the polymers. Even though Figure 1 was reproduced from the reference 49 with the wrong structure, I still suggest the authors to re-draw the polymer structures;

4. Line 130: It is worth nothing that should be noting.

Author Response

We very much appreciate your positive comments! We have now addressed all the points raised by the reviewer. Please see attached. Thanks!

Reviewer 2 Report

The review paper “Reversibility in Controlled Polymerization Systems: Recent Progress and New Opportunities” is devoted to an exciting subject due to new possibilities that can be explored with the degradability (e.g. for environmental sustainability and biomedical applications) of polymers produced through controlled radical polymerization (which molecular architecture is often tailored).

In my opinion, this paper should be interesting for the readers of the Molecules journal and recommend its publication after the addressing of the following minor issues:

1)      The authors should be aware of the IUPAC terminology for reversible-deactivation radical polymerization, before called “controlled radical polymerization” or “living radical polymerization” (Pure Appl. Chem., Vol. 82, No. 2, pp. 483–491, 2010).

2)      While the advantages concerning environmental sustainability are very clear with the reversibility of the polymerizations, practical applications in biomedical systems (e.g.) are hard to conceive in view of the aggressive conditions often used in depolymerization. I suggest that the authors improve in the manuscript the description of these new possibilities in biomedical systems (e.g.).

3)      Reversibility of polymerization in sulfur polymers is also a key concept for electrochemical applications, namely when RAFT polymerization is involved. I suggest that the authors include also the discussion of these kind of systems in the manuscript. The reference J. APPL. POLYM. SCI. 2016, DOI: 10.1002/APP.43993 should be helpful in this context.

4)      The polymer reaction engineering of the depolymerization processes should play a key role in this area if the commercial/industrial applications of these concepts are prospected. So, the prediction and design of the reversibility conditions, dynamics and final products are welcome in this context. I suggest that authors indicate to the readers of the paper extant mathematical tools that can be applied to address reversible polymerizations. In my opinion a nice overview for these tools is presented in the book: Modeling and Simulation in Polymer Reaction Engineering: A Modular Approach 1st Edition, Klaus-Dieter Hungenberg, Michael Wulkow, Wiley, 2018.

Author Response

(The authors gave the same response as above.)

Reviewer 3 Report

The authors reported an interesting review subject. However, this initial draft need to be improved in order for being considerate for publication. Mainly, the number of references and the referred work is not adequate for a review, almost all the references are in the introduction very few in the main text. The state of art have been not adequately reported. The Scheme 1 and scheme 2, do not give much information. Please if this kind of scheme are introduced it must be a reason for that. If they are added to facilitate the visualization explain it and use it in the text. Similarly, the Table 1, please develop a context and improve the explanation.

In my opinion, this review it could be highly interesting but in the present stage looks like a first draft that must be improve. A solid review must of this topic should include more references and work cited, this topic deserve it. I would be nice if you are able to improve it in order to arise readers interest. The reversible polymerization in CRP need to have a second look, considered that for all this part only 5 articles have been cited, it really need a deeper consideration.

Author Response

We very much appreciate your comments! We have now addressed all the points raised by the reviewer. Please see attached. Thanks!

Round  2

Reviewer 3 Report

Authors have slightly improved their manuscript, however, there are some issues that must be addressed before and that prevent me to recommended for its publication

Sorry, but the scheme 1 have not sense at all. On one hand, it is obvious, the readers of this Journal already know what a reversible polymerization is without a childish scheme. On the other hand, the resolution of the scheme is very poor.

Line 105. The use of the direct form must be restricted. This beginning gives an image too colloquial to the manuscript, not adjusted to a scientific article that must be avoided.

The figures all over the text must have the adequate resolution, please revise them!

When the Chen’s groups reports are described it will ease the reading of article if the times you mention “Chen group” are reduced.

Line 263, What type of NMR? Proton, carbon, COSY..?

Author Response

Authors have slightly improved their manuscript, however, there are some issues that must be addressed before and that prevent me to recommended for its publication

Sorry, but the scheme 1 have not sense at all. On one hand, it is obvious, the readers of this Journal already know what a reversible polymerization is without a childish scheme. On the other hand, the resolution of the scheme is very poor.

Response: Thanks for this point. However, we are sorry that we can not agree with the reviewer’s claim that our scheme is “childish”.  While many polymer scientists are familiar with reversible polymerization, we aim to attract wide audience to this paper, not only in the field of polymer chemistry but also in other chemistry disciplines. As the reviewer probably knows, Molecules is a world well-known journal in general chemistry. The audience volume for this journal is huge and therefore I believe those audience will benefit from those fundamental schemes with easy-to-understand information.

However, since we are planning to use scheme 1 for our TOC graphic, we decide to delete it in the manuscript. Hope this will also satisfy the reviewer.

Line 105. The use of the direct form must be restricted. This beginning gives an image too colloquial to the manuscript, not adjusted to a scientific article that must be avoided.

Response: Sorry. I tried really hard to understand this suggestion but still I am not quite sure about this. In my opinion, no rule has ever been established with regard to restricting the use of “direct form” in the writing. In the past 5 years, we have extensive experience in publishing high quality papers in well-reputed publishers including MDPI, Springer Nature, Wiley, ACS, RSC, among others. However, neither I nor my American and European colleagues have any issue with using direct form. Therefore, I regret that I can not accept this point.

The figures all over the text must have the adequate resolution, please revise them!

Response: Thanks. The figures now have good qualities satisfactory to the Journal.

When the Chen’s groups reports are described it will ease the reading of article if the times you mention “Chen group” are reduced.

Line 263, What type of NMR? Proton, carbon, COSY..?

Response: Good point. We have modified it to proton NMR. Please see page 10 for the highlighted change.
